# First Experimental Study of the Influence of Extracellular Vesicles Derived from Multipotent Stromal Cells on Osseointegration of Dental Implants

**DOI:** 10.3390/ijms22168774

**Published:** 2021-08-16

**Authors:** Igor Maiborodin, Aleksandr Shevela, Vera Matveeva, Vitaly Morozov, Michael Toder, Sergey Krasil’nikov, Alina Koryakina, Andrew Shevela, Oleg Yanushevich

**Affiliations:** 1Laboratory of Health Management Technologies, The Center of New Medical Technologies, Institute of Chemical Biology and Fundamental Medicine, The Russian Academy of Sciences, Siberian Branch, Akademika Lavrenteva Str., 8, 630090 Novosibirsk, Russia; mdshevela@gmail.com (A.S.); vam@niboch.nsc.ru (V.M.); doctor.morozov@mail.ru (V.M.); professorkrasilnikov@rambler.ru (S.K.); ashevela@mail.ru (A.S.); 2Institute of Molecular Pathology and Pathomorphology, Federal State Budget Scientific Institution “Federal Research Center of Fundamental and Translational Medicine” of the Ministry of Science and Higher Education of the Russian Federation, Akademika Timakova Str., 2, 630117 Novosibirsk, Russia; 3International Center of Dental Implantology iDent, Sibrevkoma Str., 9b, 630007 Novosibirsk, Russia; drtoder@gmail.com (M.T.); alinakoryakina@gmail.com (A.K.); 4Moscow State University of Medicine and Dentistry, The Ministry of Healthcare of the Russian Federation, Delegatskaya Str., 20, p. 1, 127473 Moscow, Russia; mail@msmsu.ru

**Keywords:** bone tissue, extracellular vesicles, bone tissue density, dental implantation, implant osseointegration

## Abstract

Herein, the aim was to study the state of the bone tissue adjacent to dental implants after the use of extracellular vesicles derived from multipotent stromal cells (MSC EVs) of bone marrow origin in the experiment. In compliance with the rules of asepsis and antiseptics under general intravenous anesthesia with propofol, the screw dental implants were installed in the proximal condyles of the tibia of outbred rabbits without and with preliminary introduction of 19.2 μg MSC EVs into each bone tissue defect. In 3, 7, and 10 days after the operation, the density of bone tissue adjacent to different parts of the implant using an X-ray unit with densitometer was measured. In addition, the histological examinations of the bone site with the hole from the removed device and the soft tissues from the surface of the proximal tibial condyle in the area of intra-bone implants were made. It was found out that 3 days after implantation with the use of MSC EVs, the bone density was statistically significantly higher by 47.2% than after the same implantation, but without the injection of MSC EVs. It is possible that as a result of the immunomodulatory action of MSC EVs, the activity of inflammation decreases, and, respectively, the degree of vasodilation in bones and leukocyte infiltration of the soft tissues are lower, in comparison with the surgery performed in the control group. The bone fragments formed during implantation are mainly consolidated with each other and with the regenerating bone. Day 10 demonstrated that all animals with the use of MSC EVs had almost complete fusion of the screw device with the bone tissue, whereas after the operation without the application of MSC EVs, the heterogeneous histologic pattern was observed: From almost complete osseointegration of the implant to the absolute absence of contact between the foreign body and the new formed bone. Therefore, the use of MSC EVs during the introduction of dental implants into the proximal condyle of the tibia of rabbits contributes to an increase of the bone tissue density near the device after 3 days and to the achievement of consistently successful osseointegration of implants 10 days after the surgery.

## 1. Introduction

All over the world, the number and volume of surgery associated with the implantation of products made from artificial materials, including metals, are increasing [1]. Reconstruction of the patient’s tissues should be considered not only from an aesthetic point of view, achieving a cosmetic result, restoring lost functions, and ensuring adequate hygiene are of great importance.

Despite the many general reactions of the body, which perceives the implant as a foreign body, individual effects are often associated with the properties of the materials from which the implant is made [2]. Modification of the surface of products is important to improve osseointegration [1,3,4].

Cell technologies have certain prospects for modification of the surface of products and improving the results of intraosseous implantation. Cells influence each other and exchange functional proteins and genetic material through the secretion of exosomes/extracellular vesicles (EVs), which can be used for influence on tissue regeneration [5]. EVs, which are very small membrane formations with the diameter of 40 to 100 nm, formed from endosomal multivesicular bodies through fusion with the surface cytoplasmic cell membrane, are produced by various cells and contain signaling molecules that transmit precise information for target cells from the well-defined cell. That is, EVs of certain cells may be new tools for regenerative therapy [6,7].

Most likely, the mechanism of influence on repair is the transport of microRNA by EVs. Moreover, EVs interact with matrix proteins such as fibronectin and type I collagen, allowing them to be used as biomaterials [5,8,9]. Cytokines and EVs secreted by MSCs of bone marrow origin are the main factor in the restoration of damaged tissues. Regulatory releases of MSCs can act as the critical modulator of repair [10].

EVs can induce and influence the differentiation of multipotent stromal/stem cells (MSCs) in a certain direction, including the osteoblastic one. It was found that releases derived from MSCs of the human umbilical cord lead to the onset of osteogenic differentiation of MSCs isolated from the bone marrow and promote bone tissue repair [11]. EVs produced by adipose tissue MSCs promote proliferation and osteogenic differentiation of primary human osteoblast culture [12]. In vivo, MSC EVs strongly stimulated bone regeneration and angiogenesis in critical-sized calvarial bone defects in ovariectomized rats. The effect of MSC EVs increased with a growth in their concentration [13].

To study the results of application of EVs derived from MSCs of bone marrow origin of rat for the regeneration of bone tissues, the defect (2 mm diameter and 4 mm depth) was created in the proximal condyles of the outbred rabbit tibia. On the left limb, the bone defect was filled with saline, on the right, the 50 μg MSC EVs were introduced into the defect. By the 12th day, all of the control rabbits had retained a defect in the bone tissue along with forming bone structures and scar in the border with intact areas. In most cases after the MSC EVs introduction, no bone damage was found. The scar was thin with ordered structures of the intercellular matrix. Twelve days after the application of Vybrant^®^ CM-Dil-labeled MSC EVs, in the periosteum and in adjacent bone marrow vessels of bone tissue, the single, very small, dust-like objects were found fluorescent in red in the time of rhodamine filter use. Sometimes, the clear red tint of inclusions was observed in large cellular elements—macrophages. By the 21st day on the right (experimental) limb, in four cases out of five, only scar structures were found at the site of the bone tissue defect, while on the left (control) limb—only two cases were left out of five. It was concluded that the MSC EV application for the regeneration of bone defect revealed the faster healing, the increase of frequency of successful regeneration of the damaged bone, and the formation of a less rough bone callus [14].

Probably, due to MSCs or their EVs, which have an immunomodulatory effect [15,16,17], it is theoretically possible to reduce the severity of acute (during implantation) and chronic (accompanying the presence of a foreign body in tissues) inflammatory process. The effect of EVs is more similar to the effects of MSCs [18].

In view of the above, the following aim of the study was set: To study the state of the bone tissue adjacent to dental implants after the use of mesenchymal MSC EVs in the experiment.

## 2. Results

During densitometry and combining the data obtained from all five measurement points around the implanted device, it was found that 3 days after implantation with the use of MSC EVs, the bone density was statistically significantly higher by 47.2% than after the same implantation, but without the application of MSC EVs (Table 1 and Table 2; Figure 1a,b, Figure 2a,b, and Figure 3a).

The density of bone tissue around the implant after surgery without MSC EVs on 7 and 10 days increased by 63.5 and 56.7%, respectively, relative to the period of 3 days (Table 1; Figure 3a). There was no reliable dynamics of bone density after implantation under conditions of preliminary introduction of MSC EVs depending on the date of observation (Table 2; Figure 3a).

On the 3rd day after implantation without the use of MSC EVs, there was a lot of bone detritus—non-viable bone fragments of various sizes and shapes—between the implants and the structures of the intact bone (Figure 1c,d). By day 7, the bone formation near the implant was observed only in small areas. Basically, the implant was bordered with structures of progressive growths of loose fibrous connective tissue with inflammatory infiltration and many separately located non-viable bone fragments. In some places, there was consolidation of viable bone fragments into the trabeculae of bone, and in others—lysis of necrotic bone fragments.

Ten days after surgery without the use of MSC EVs, one case demonstrated a complete osseointegration of the implanted device, which was in contact with the bone tissue from all sides. In another case, the implant was mainly surrounded by cancellous bone, and only in small areas it was bordered with structures of loose fibrous connective tissue (Figure 1e). The 3rd rabbit had the entire implant in loose fibrous connective tissue with a large number of non-viable encapsulated bone fragments (Figure 1f). For this period, the values of the ordered series of the length of metal contact with bone tissue, compiled from data from all animals, were in the range from 0 to 90% of the length of the entire device surface on the sections, which led to a high standard deviation of the obtained mean and the absence of statistically significant differences from the results of the study on previous dates (Table 1; Figure 3b).

As a difference of implantation with the injection of MSC EVs, it can be noted that on the 3rd day, the bone detritus was grouped, compressed and, accordingly, occupied a slightly smaller volume (Figure 2c). By the 7th day, the screw device was surrounded by various types of fibrous connective tissue, which contained many new formed trabeculae of bone. In addition, bone fragments were located everywhere near the foreign body, mainly in small groups, and an intensive formation of new formed bone tissue was recorded next to the bone fragments (Figure 2d). This may be due to both the consolidation of viable split bone particles between themselves and the formation of the new formed bone from the material of lysed “old” fragments.

On the 10th day after surgery with the preliminary introduction of MSC EVs, all the animals demonstrated that only in very insignificant places the implanted device was delimited from the bone by loose fibrous connective tissue, where there were large fragments of still unlyzed bone detritus. Throughout the rest of the length, the foreign body was in close contact with the bone structures, and all the debris was integrated into the growing structures of new formed bone tissue (Figure 2e,f). On the 10th day after using MSC EVs, the values of the ordered series of the length of metal-to-bone contact were stable and were in the range between 60 and 90%. As a result, after 10 days, the percentage of the foreign body contact with bone tissue became 4.8 times higher than on the 7th day (Table 2; Figure 3b). However, at the same time, there were no significant differences from the data obtained during implantation without the use of MSC EVs, due to the already noted high value of the standard deviation in the latter case (Table 1 and Table 2; Figure 3b).

The relative area of blood vessels on the bone tissue section near the dental implant after surgery without the application of MSC EVs was practically constant throughout the observation period and ranged from 9 to 15% of the entire tissue section (Table 1; Figure 1c,d and Figure 3c).

The density of vessels after implantation with preliminary introduction of MSC EVs after 3 days was lower by 95.7 and 94.1%, respectively, than for the period of 7 and 10 days (Table 2; Figure 3c). At the same time, the value of this indicator on the 3rd day after using MSC EVs was 85.9% less, relative to the results of standard implantation for the same period (Table 1 and Table 2; Figure 1c,d, Figure 2c, and Figure 3c).

The numerical density of all cells in the soft tissue sections from the surface of the proximal tibial condyle in the area of the rabbit’s intraosseous implants of the control group after 10 days decreased by 97.7% and 2.1 times compared with the data by 3 and 7 days of the experiment, respectively. The percentage of lymphocytes on day 10 became statistically significantly lower by 52.4% than 3 days after implantation. The absolute number of lymphocytes after 10 days decreased by 3 and 3.2 times relative to the state on days 3 and 7 after surgery, respectively. The relative content of neutrophils after 10 days decreased by 4.8 and 2.6 times compared with the results on the 3rd and 7th days of observation, respectively. At the same time, by the 7th day, the content of these cells became lower by 84.4% than on the 3rd day. The number of neutrophilic leukocytes per unit of section area after 7 and 10 days decreased by 72.4% and 9.4 times, respectively, relative to the state on day 3 after surgery. By day 10, the value of this indicator was 5.5 times lower than on day 7. The percentage of macrophages among all cells increased by day 7 and was statistically significantly higher by 77.7% compared to the data on day 3. However, further, the value of this indicator decreased and again did not differ from the results on the 3rd day. The numerical density of macrophages after 10 days was 3.2 times less than on the 7th day of observation (Table 1; Figure 3d–f).

The numerical density of all cells in the tissues on the surface of the proximal tibial condyle next to the intraosseous implants in rabbits using MSC EVs after 10 days decreased by 71.7 and 63.4% compared with the data by 3 and 7 days of observation, respectively. The absolute number of lymphocytes after 10 days decreased by 65.3 and 72.5% relative to the state on days 3 and 7 after surgery, respectively. The relative content of neutrophils after 7 and 10 days decreased by 94.1% and 2.7 times, respectively, compared with the results of the experiment on day 3. The number of neutrophilic leukocytes per unit section area after 7 and 10 days decreased by 2.1 and 4.7 times, respectively, relative to the state on the 3rd day after surgery. By the 10th day, the value of this indicator was 2.3 times lower than on the 7th day. The percentage of macrophages among all cells on days 7 and 10 was 2.4 and 2.5 times higher, respectively, compared to the data on day 3. The numerical density of macrophages after 7 days was 2.3 times higher than on the 3rd day of observation. However, further, by day 10, the value of this indicator decreased by 57.4% and did not differ statistically significantly from the results on day 3 (Table 2; Figure 3d–f).

When comparing the results of studying the number and ratio of cells in soft tissues next to a metal implant inserted into the tibia between groups of experimental animals without and with MSC EV administration, it was found that after using MSC EVs, the numerical density of all cells was lower by 3 days by 40.3%; the percentage and absolute number of lymphocytes—by 48.8% and 2.1 times, respectively; the content of neutrophils per unit area of the tissue section—by 57%. By day 7, as a result of using MSC EVs, the absolute amount of all cells was lower by 59.3%, and neutrophilic leukocytes—by 88%. The introduction of MSC EVs led to the fact that on the 10th day the relative content of macrophages was higher by 86.1% (Table 1 and Table 2; Figure 3d–f).

## 3. Discussion

The fact that 3 days after surgery with the use of MSC EVs, the density of bone tissue around the implant is statistically significantly higher than after the same surgery, but without MSC EVs, is the basis for the assumption about the effect of MSC EVs on the repair of bone tissue around dental implants.

In addition, after implantation without MSC EVs, there is a rapid and statistically significant increase in bone density. In the time of MSC EV use, there were no statistically significant changes within 10 days of observation. This, apparently, is due to the fact that already on the 3rd day after the operation, the bone density of the experimental animals is higher than in the control ones, and it is significant. A further increase in bone density near the implant due to the already initially high values is less intense due to the primary difference.

Tissue damage during the preparation and the implantation procedure itself is accompanied by the formation of detritus. Even the use of water cooling during the implantation process does not lead to the complete removal of small bone fragments, which, when the screw device is introduced, are pushed back into the surrounding tissues and are compressed there. The prolonged presence of debris at the site of implantation, a corresponding prolongation of the inflammatory process, and a delay in tissue regeneration can impair the engraftment of implanted devices.

It is most likely that the high rates of tissue vascularization on the 3rd day after implantation without the use of MSC EVs are due to the inflammatory process caused by surgical bone injury. Moreover, one of the causes of inflammation is the presence of a large volume of bone detritus in the tissues next to the implant. With an inflammatory reaction, many biologically active substances appear in the tissues that contribute to stasis, thrombosis, and the corresponding expansion of blood vessels, which is a physiological reaction to damage that prevents the spread of antigens and toxins from the site of injury throughout the body.

After implantation without MSC EVs the worsening hyperemia moves apart the bone fragments, pushes them away from each other, while the feeding vessels are damaged or clamped down. This leads to the necrotization of a larger number of bone fragments with the need for their lysis, which, in turn, prolongs inflammation and postpones the integration time of the implant. Then, the activity of the inflammatory process subsides and is supported only by the presence of still unlyzed bone fragments. The volume of the enlarged vessels is normalized, but the tissues are characterized with angiogenesis, which is necessary both for repair with the formation of granulations and for osteogenesis (angiogenic osteogenesis) and osseointegration of the implants. Since, until the end of the observation, non-viable bone fragments remain in the tissues, and there was no complete integration of the implanted device, the phenomena of inflammatory hyperemia and angiogenic osteogenesis are also present 10 days after the operation.

As a result of the MSC EV introduction, the activity of inflammation decreases [16,17] and, respectively, the degree of vasodilation is lower. Then, osteogenesis begins according to the angiogenic type, and the volumetric density of vessels at the cut rises by the 7th day and remains at this level until the end of the experiment.

As a result of a decrease in the intensity of the inflammatory process by MSC EVs the grouped bone fragments next to the implant are not subjected to lysis and damage by exocyte enzymes of leukocytes [19,20], but are consolidated, set with each other and with intact bone structures. At the same time, there are no disturbances in the blood supply of these bone particles, since the feeding vessels are not clamped down by either hyperemic vessels or edema, which is also caused by blockage of blood flow and hyperemia or inflammatory infiltrate. It is possible that these bone particles, in addition to consolidation, can even serve as starting points of bone regeneration, a source of osteoblasts, where the formation of new osteons and trabeculae begins.

It is possible that the compression of split bone fragments during screwing of the device under conditions of less pronounced hyperemia and edema is the main reason for the increase in bone tissue density detected on the 3rd day after implantation with the introduction of MSC EVs. The angiogenesis that develops by the 7th day leads to a decrease in the density of structures, but for this period there is no separation of the already tightly fastened fibrin and the bone fragments that have begun consolidation. After surgery without the use of MSC EVs, a compression of the bone particles occurs first, however, rapidly developing hyperemia and accompanying edema immediately push these bone fragments apart, thereby reducing the density of tissues next to the implant after 3 days.

The relative stability of the morphometric data in osseointegration study after implantation with the introduction of MSC EVs, probably, is also associated with their immunomodulatory effect [15,16,17]. MSC EVs control of inflammation activity [16,17], less severity of vasodilation and edema, apparently, led to the fact that split bone fragments remain in a compacted state, as soon as they are compressed by screwing in an implant. The process of osteogenesis with the use of MSC EVs is less dependent on the number of split bone fragments formed during implantation, since, if there are many of them, they are consolidated and form the basis for the growth of new formed bone. In addition, if there are few—they quickly lyse and do not interfere with the formation of bone both from intact trabeculae and from blood vessels (by the type of angiogenic osteogenesis). A much smaller volume of necrotic bone tissue undergoes lysis and, accordingly, this lysis ends faster, especially under conditions of macrophages activation by MSC EVs [16].

The dynamics of changes in the number and ratio of leukocytes in the soft tissue sections from the surface of the proximal tibial condyle in the area of intra-bone implants in control animals corresponds to the course of the wound process. As detritus, which has antigenic properties, is eliminated from the surgical site, the content of all cells, and, in particular, lymphocytes and neutrophils, decreases in the tissues near the implant. Attention is drawn to the increase in the number of macrophages by the 7th day, with subsequent normalization. It is possible that such changes are associated with the development of granulomatous inflammation for the elimination of large fragments of detritus and necrotic bone particles. As the lysis and elimination of detritus and bone fragments, as well as the gradual completion of sclerosis and scar formation, by the end of the observation, the content of macrophages decreases.

It can be noted that the use of MSC EVs led to a smaller number of leukocytes in the soft tissues from the surface of the proximal tibial condyle already on the 3rd day after implantation and to a less pronounced decrease in their number further, in comparison with the surgery performed in the control group. These results are best explained from the standpoint of the well-known immunomodulatory effect of MSC EVs [15,16,17], which results in suppression of lymphocyte [15] and neutrophil [16,17,21] activity, and the severity of inflammation is also suppressed [16,17]. It is necessary to pay attention to the higher number of macrophages on the 10th day after the operation with the use of MSC EVs. This may indicate both the ongoing granulomatous inflammation caused by the need to lysis of bone and other dense fragments of detritus, and the possible compensation by macrophages of the functions of other leukocytes suppressed by MSC EVs [15,16,17,21].

## 4. Materials and Methods

The research is based on findings obtained when studying the samples of tibia of outbred rabbits of both sexes weighing 3–4 kg at various times after the injection of MSC EVs into an artificially created defect of the proximal condyle of this bone, with subsequent installation of screw titanium implants. The manipulations did not cause pain to animals and were carried out in compliance with the Russian legislation: GOST 33215-2014 (guidelines for accommodation and care of laboratory animals. Rules for equipment of premises and organization of procedures) and GOST 33216-2014 (guidelines for accommodation and care of laboratory animals. Rules for the accommodation and care of laboratory rodents and rabbits). The work is approved by the Local Committee on Biomedical Ethics (24 October 2014, decision No. 18) at the Center of New Medical Technologies, Institute of Chemical Biology and Fundamental Medicine, The Russian Academy of Sciences, Siberian Branch.

### 4.1. Preparation, Cultivation, and Characteristics of MSCs, and Isolation of MSC EVs

MSCs were obtained from the bone marrow of a male Wag inbred line rat weighing 180 g and aged 6 months and then were characterized and cultured as described in our previous works [14,22,23,24]. Isolated MSCs expressed characteristic MSC marker CD90 and did not express the hematopoietic marker CD45 (Appendix A; Appendix A).

The culture nutrient medium containing fetal bovine serum and used for MSC growth was preliminarily subjected to complete gradient centrifugation and purified from its own EVs. At the stage of stationary growth of a stable culture of the 3rd MSC passage, when the confluence of the cell monolayer reached 80–90%, a conditioned medium was collected from which MSC EVs were isolated as recommended in the literature [14,15,25,26,27]. To remove cells, cell debris, apoptotic bodies, and large vesicles, the conditioned medium was centrifuged sequentially: 10 min in the case of 300× *g*, 10 min in the case of 2000× *g*, and 30 min in the case of 12,000× *g*. EVs were precipitated by centrifuging the supernatant for 2 h in the case of 100,000× *g* and resuspended in saline with phosphate buffer.

MSC EVs were analyzed by electron transmission microscopy and flow cytometry. Relevant markers for EVs secreting MSCs, tetraspanin CD63, were detectable [28,29]. When preparing samples for the research, the objects were sorbed onto a copper grid covered with a formvar film for 1 min and contrasted with a 2% phosphotungstic acid solution for 10 s. The grids were studied in the transmission mode of a Jem 1400 electron microscope (Jeol, Tokyo, Japan), and images were obtained using a Veleta digital camera (Olympus Corporation, Tokyo, Japan). The particle size was determined in 5–8 randomly selected fields of view at 60,000 times magnification using the iTEM software package (Olympus Corporation, Japan). More than 90% of the objects had a diameter of 70–90 microns and a three-layer membrane.

The isolated extracellular particles were adsorbed onto 4 μm aldehyde/sulfate latex particles (Invitrogen, 37304) [30] and analyzed on a NovoCyte™ cytometer using its software (ACEA Biosciences Inc., San Diego, CA, USA). To detect EVs, antibodies specific to marker-specific exosome proteins (Bio-Rad, MCA4754F, FITC mouse IgG1, k) were used. An isotype control (Bio-Rad, MCA1209, isotype control, FITC mouse IgG1, k) was used to measure nonspecific sorption. In each experiment, no less than 30,000 events were counted. The amount of MSC EVs was determined by the protein content in the precipitate using a commercial Qubit protein assay kit (Thermo Fisher Scientific, Waltham, MA, USA) and a Qubit^®^ 3.0 fluorometer.

### 4.2. Introduction of MSC EVs into a Bone Defect

Surgical intervention was performed in compliance with all the rules of asepsis and antiseptics under general intravenous anesthesia with propofol. In both proximal tibial condyles of rabbits, standardized 4 mm holes were created with a 2 mm dental bur and cooled by sterile saline solution [14,27,31].

Next, an insulin syringe was used to fill the bone defect with physiological saline prepared in phosphate buffer (pH = 7.3) (control, 9 rabbits, 3 pieces for each experiment date) [32] or 19.2 μg of MSC EVs in saline solution injected for each limb (experiment, 10 animals, 4 rabbits for a period of 3 days, and 3 animals for each observation period). The MSC EVs dose was selected based on the average dose recommended by other researchers: 10–20 μg/mL [11]; 0.6 μg, 5 μg or 50 μg [33]; 50 μg for the same bone tissue defect of the proximal tibial condyle [14,27]; 100 μg immediately after surgery and weekly for 12 weeks [32]. After 10–20 s, titanium screw implants (catalog number IS 358; 3.5 × 8 mm with a rough surface; 3S, Israel) were inserted with a stable primary fixation up to 30 Ncm, and the surgical wound was sutured layer by layer without tension [14,27,31].

After 3, 7, and 10 days, the animals were sacrificed by dislocation of the cervical vertebrae. Each group consisted of 3–4 animals, 19 animals in total.

### 4.3. X-Ray, Morphological, and Morphometric Research Methods

After euthanasia, the limbs of rabbits with implants were fixed in 4% paraformaldehyde solution in phosphate buffer (pH = 7.4) for at least 7 days, then, carefully not to displace the implanted devices, the shins were isolated, and the skin and soft tissues were removed.

The shins with implants were studied using the ORTHOPHOS XG device: Voltage on the X-ray tube—69 kV; current—15 mA; duration—13,962 ms. The images were obtained using the Sidexis 4 Viewer Version 4.3 software; 3D modeling and densitometry—with the help of GALAXIS Basis/GALILEOS Implant Viewer. The density of bone tissue adjacent to the apex of the implant was measured, to its neck (2 measurements on the right and left) and at the level of 3–4 threads from the apex (also 2 measurements), the results of densitometry are expressed in optical units, where “0” is the density of the vacuum, and 100% or 4095 units is the state of the dense metal. All the equipment and software are manufactured by ©Sirona Dental Systems GmbH, Bensheim, Germany.

For the morphological examination, the implants were removed from the fixed tibia. Since the implants were removed from the fixed tissues, the damage associated with the extraction of the implants was clearly distinguishable from the intravital bone changes—the reaction of the bone tissue to the insertion and the presence of a foreign body. The fragment with the hole from the screw device was cut out and decalcified with Biodec R solution (Bio Optica Milano, Milano, Italy). The decalcified bone and the soft tissues from the surface of the proximal tibial condyle in the area of intra-bone implants was dehydrated in the Isoprep reagent (BioVitrum, St. Petersburg, Russia), clarified in xylene and embedded in the histoplast. The block of the histoplast with bone material during the production of sections was oriented in such a way that the maximum area of the histological section was either parallel to the hole from the removed implant (bone sections with gear marks from the thread faces) or across its axis (round hole on the sections). The bone was sampled in serial sections with the thickness of 5–7 μm for the entire volume of the extracted implant (this made it possible to study the entire area of contact between the bone and the implant), which, after staining with hematoxylin and eosin, were examined at magnification up to 1200 times on an Axioimager M1 light microscope (Carl Zeiss, Jena, Germany). Inflammatory cell infiltration has only been studied in soft tissue from the surface of the tibial condyle. Leukocyte infiltration in bone tissue and in tissues immediately adjacent to the implant has not been investigated, since long-term decalcification strongly changes the morphology of cells.

To obtain the necessary morphometric data, the images were measured using the digital video camera of this microscope with the help of the Axiovision morphometry software package (Carl Zeiss, Jena, Germany). The vascularization of tissues near the implanted devices was determined to double the depth of their screw thread. The length of the contact implant profile with the bone tissue was set as a percentage of the total length of the contact profile of the screw thread with the tissues, in addition the curvimeter was used. To determine the severity of inflammatory infiltration (the total number of cells and separately the number of lymphocytes, neutrophils, and macrophages), the images obtained with a digital video camera of a microscope were measured on a computer screen using the software of the Axiovision morphological module (Carl Zeiss, Jena, Germany). When using a lens with magnification of 40×, the area of the rectangular image was 8.7 × 10^4^ μm (sides 350 × 250 μm). Each section was measured 3–5 times at different areas according to the instructions, which it is enough to study 3 sections from each object to obtain statistically reliable results [34].

During statistical processing of the obtained data, the arithmetic mean and standard deviation were determined. The significance of the difference between the compared mean values was determined on the basis of the Student’s test. The difference between the compared series with the confidence level of 95% and higher was considered significant. The calculations took into account that the distribution of the studied characters was close to normal.

## 5. Conclusions

It can be concluded that the use of MSC EVs during the administration of dental implants into the proximal condyle of the tibia of rabbits contributes to an increase in the density of bone tissue near the device 3 days after surgery. It is possible that as a result of the immunomodulatory action of MSC EVs, the activity of inflammation decreases, and, respectively, the degree of vasodilation in bones and leukocyte infiltration of the soft tissues are lower, in comparison with the surgery performed in the control group. The bone fragments formed during implantation are mainly consolidated with each other and with the regenerating bone. When using MSC EVs, on the 10th day all the animals demonstrated almost complete fusion of the screw device with bone tissue, whereas after surgery without applying MSC EVs, a heterogeneous histologic pattern was observed: From almost complete osseointegration of the implant to the absolute absence of contact between a foreign body and new formed bone.

## Figures and Tables

**Figure 1 ijms-22-08774-f001:**
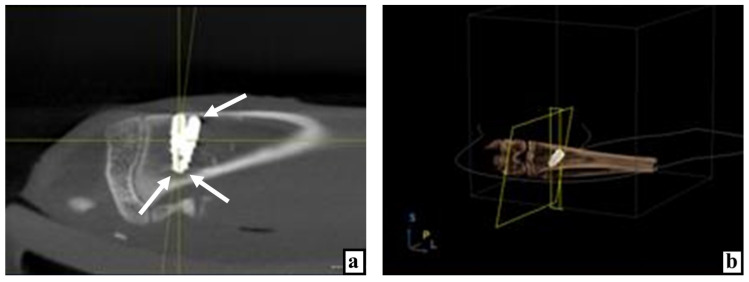
Results of studying the bone tissue of the proximal tibial condyle in control rabbits at various times after the introduction of dental implants. (**a**) X-ray image of the implant in the tibia 3 days after surgery, arrows indicate areas of reduced bone density. (**b**) Computer 3D modeling of the implant position in the bone on the 3rd day after surgery. (**c**) On the 3rd day after implantation, there are extensive hemorrhages and a large volume of split bone fragments in the cancellous bone, staining with hematoxylin and eosin. (**d**) Hemorrhages, vascular distention, and congestion (arrows) near and among the split bone particles 3 days after implant placement, stained with hematoxylin and eosin. (**e**) On the 10th day, the implant mostly borders with the bone tissue and only in small areas it is adjoined by loose fibrous connective tissue (arrows) with a significant number of hyperemic wide thin-walled vessels, stained with hematoxylin and eosin. (**f**) After 10 days the implant is delimited from the bone throughout its entire length by connective tissue with wide hyperemic vessels, stained with hematoxylin and eosin.

**Figure 2 ijms-22-08774-f002:**
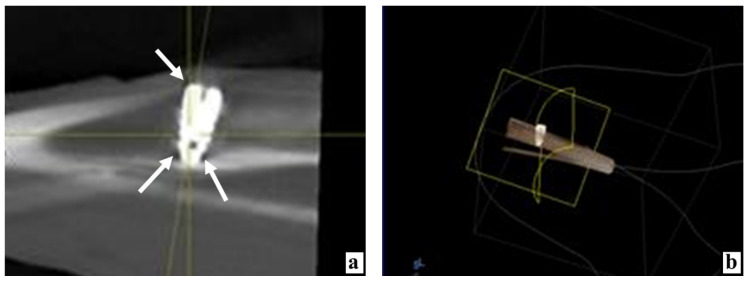
The state of the animal tibia at various times after the introduction of dental implants into the proximal condyle with the preliminary injection of MSC EVs. (**a**) By the 3rd day after the operation, according to the X-ray examination, the tissue rarefaction near the implant neck is insignificant (arrow), in the apex area—only along its sides (arrows). (**b**) Computer 3D modeling of the device position in the bone 3 days after implantation. (**c**) Three days after the operation, the fragments of bone detritus adhere tightly to each other, between them there are practically no corpuscles and blood vessels, stained with hematoxylin and eosin. (**d**) On the 7th day after implantation, next to the split parts of the bone tissue infiltrated by large cells, there are structures of new bone, possibly formed from these bone fragments, stained with hematoxylin and eosin. (**e**) By the 10th day, only for a short distance, the implant borders with the connective tissue (arrows), stained with hematoxylin and eosin. (**f**) Partial lysis, consolidation with each other, and ingrowth of split bone fragments into new forming bone structures 10 days after surgery, stained with hematoxylin and eosin.

**Figure 3 ijms-22-08774-f003:**
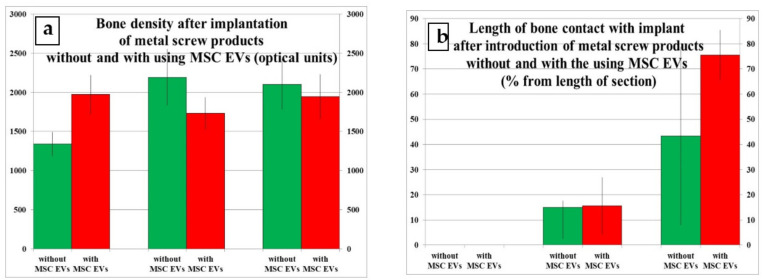
Results of comparing bone density, vascularization, and leukocyte infiltration in tissues of the rabbit’s proximal tibial condyle at different times after the introduction of screw implants without and with the use of cells. (**a**) Bone density (optical units) after implantation of metal screw products without and with using MSC EVs. (**b**) Length of bone contact with implant (% from length of section) after introduction of metal screw products without and with the using MSC EVs. (**c**) Relative area of blood vessels (% from section area) in section. (**d**) Numerical density of all cells (per 105 μm^2^ of the section area) in the soft tissues from surface of the condyle. (**e**) Numerical density of lymphocytes (per 105 μm^2^ of the section area) in the soft tissues from surface of the condyle. (**f**) Numerical density of neutrophils (per 105 μm^2^ of the section area) in the soft tissues from surface of the condyle.

**Table 1 ijms-22-08774-t001:** Condition of the rabbit’s tissues from the proximal tibia condyle near the dental implants at control animals (S ± σ).

Parameter	Time after Implantation		
	3 Days	7 Days	10 Days
Total bone density near the implant (optical units)	1341 ± 153	2193 ± 356 ^#^	2101 ± 323 ^#^
Length of bone contact with implant (L_L_)	-	15 ± 12,5	43.3 ± 35.4
Blood vessels (A_A_)	11.3 ± 2.24	12.1 ± 2.09	12.2 ± 1.79
Numerical density of all cells (N_A_)	174 ± 20.1	188 ± 33.5	88 ± 13.9 ^#,$^
Lymphocytes (%) (N_A_)	19.2 ± 2.77 33.5 ± 6.23	18.4 ± 3.28 34.8 ± 10.1	12.6 ± 1.33 ^#^ 11 ± 1.57 ^#,$^
Neutrophils (%) (N_A_)	20.1 ± 2.93 35 ± 5.52	10.9 ± 1.45 ^#^ 20.3 ± 3.37 ^#^	4.22 ± 0.833 ^#,$^ 3.72 ± 1.05 ^#,$^
Macrophages (%) (N_A_)	8.44 ± 1.59 14.9 ± 4.04	15 ± 2.4 ^#^ 28.4 ± 7.81	10.1 ± 1.27 8.9 ± 1.95 ^$^

Note: L_L_—the relative length of structure in section (% from length of section); A_A_—relative area of structure in section (% from section area); N_A_—the numerical density of cells per 10^5^ μm^2^ of the section area; ^#^—values significantly differing from the corresponding on the 3rd day after the surgery (*p* ≤ 0.05); ^$^—values significantly differing from the corresponding on the 7th day after the surgery (*p* ≤ 0.05).

**Table 2 ijms-22-08774-t002:** Condition of tissues from the proximal tibia condyle near the dental implants after surgery using MSC EVs (S ± σ).

Parameter	Time after Implantation		
	3 Days	7 Days	10 Days
Total bone density near the implant (optical units)	1974 ± 248 *	1732 ± 204	1948 ± 283
Length of bone contact with implant (L_L_)	-	15.6 ± 11.3	75.6 ± 9.82 ^#,$^
Blood vessels (A_A_)	6.08 ± 1.08 *	11.9 ± 2.2 ^#^	11.8 ± 1.99 ^#^
Numerical density of all cells (N_A_)	124 ± 11.6 *	118 ± 9.71 *	72.2 ± 13.9 ^#,$^
Lymphocytes (%) (N_A_)	12.9 ± 1.24 * 16 ± 2.03 *	14.2 ± 1.48 16.7 ± 2.09	13.6 ± 1.42 9.68 ± 1.51 ^#,$^
Neutrophils (%) (N_A_)	17.9 ± 1.62 22.3 ± 2.98 *	9.22 ± 1.2 ^#^ 10.8 ± 1.35 ^#,^*	6.67 ± 1 ^#^ 4.74 ± 0.795 ^#,$^
Macrophages (%) (N_A_)	7.5 ± 0.798 9.34 ± 1.56	18.2 ± 1.09 ^#^ 21.4 ± 1.67 ^#^	18.8 ± 1.64 ^#,^* 13.6 ± 3 ^$^

Note: L_L_—the relative length of structure in section (% from length of section); A_A_—relative area of structure in section (% from section area); N_A_—the numerical density of cells per 10^5^ μm^2^ of the section area; ^#^—values significantly differing from the corresponding on the 3rd day after the surgery (*p* ≤ 0.05); ^$^—values significantly differing from the corresponding on the 7th day after the surgery (*p* ≤ 0.05); *—values significantly different from the corresponding ones after implantation without MSC EVs (*p* ≤ 0.05).

## Data Availability

The data presented in this study are available on request from the corresponding author.

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
