# Peer review of "First Experimental Study of the Influence of Extracellular Vesicles Derived from Multipotent Stromal Cells on Osseointegration of Dental Implants"

_ijms, 2021, doi:10.3390/ijms22168774_

Round 1
Reviewer 1 Report
The manuscript and the study it describes does not reach the standards that are required for an international, high-ranking publication in the field. Among the shortcomings are the following.
The quality of English used in the manuscript is so low, that over large expanses of the text, it is not really possible to understand what the authors meant or this at least remains rather doubtful. It is not the readers job to guess, which message the manuscript intends to convey.
The introduction is only 12 lines long and hardly provides a summary of the state of research about the use of extracellular vesicles in bone regeneration.
Most importantly the methods applied are not adequate to the aim of the study.
First of all, the use of paraffin histology and the fact that the implant is removed from the tissue during preparation, does not allow a certain and unambiguous study of the implant interface where the osseointegration takes place. The use of undecalcified thin ground sections would have provided reliable results and the most important parameter in the evaluation of osseointegration, the bone-to-implant contact, could have been measured reliably. When the implant is removed, bone can adhere to the surface and thereby distort the results.
There are also severe doubts that the surgical procedures were applied properly. Implants should be installed in such a way, that they are primarily located in the cortical bone of the tibia with its coronal part. Figure 1b clearly shows that no primary stability was achieved and the implant had slipped into the marrow space. Osseointegration is very different there from the regions directly adjacent to the cortical bone.
The main result of the study as it is presented, appears to be the higher bone density at 3 days. At this time point usually no or very little bone formation has taken place. The dominating process at three days is resorption of fractured bone debris produced by the drilling process. The higher bone density in the treatment group is very likely a result of more bone debris present. That means less resorption? It hardly proves that better osseointegration has taken place as a result of treatment with extracellular vesicles. And if so why is there no higher bone density after 7 and 10 days?
The interpretation of the findings is by far too optimistic, appears to be one-sided, uncritical and biased in the direction of the hoped for results.
Author Response
I am grateful to the Dear Reviewer for the work and time spent in studying our manuscript. Regarding the comments and recommendations made:
The quality of English used in the manuscript is so low, that over large expanses of the text, it is not really possible to understand what the authors meant or this at least remains rather doubtful. It is not the readers job to guess, which message the manuscript intends to convey.
I can only apologize for the quality of the English language. But the translation was made by a certified translation agency, and then revised by a US citizen.
The introduction is only 12 lines long and hardly provides a summary of the state of research about the use of extracellular vesicles in bone regeneration.
Fixed. The "Introduction" has been expanded.
Most importantly the methods applied are not adequate to the aim of the study.
First of all, the use of paraffin histology and the fact that the implant is removed from the tissue during preparation, does not allow a certain and unambiguous study of the implant interface where the osseointegration takes place. The use of undecalcified thin ground sections would have provided reliable results and the most important parameter in the evaluation of osseointegration, the bone-to-implant contact, could have been measured reliably. When the implant is removed, bone can adhere to the surface and thereby distort the results.
I cannot agree with the opinion of the Dear Reviewer that we should not have removed a metal implant during preparatory procedures. A metal product left in the bone would not make it possible to cut histological sections from tissues embedded in paraffin or polished sections from non-decalcified bone. In addition, since the implants were removed from the fixed tissues, the damage associated with the extraction of the implants was clearly distinguishable from the intravital bone changes - the reaction of the bone tissue to the insertion and the presence of a foreign body.
We made serial paraffin sections from decalcified tissues, which allowed us to study the entire area of ​​contact between the bone and the implant. Whereas the study of polished non-decalcified sections would make it possible to investigate the contact surface of the bone with the metal only in a limited area, on a random cut.
Clarifications have been added to the "Materials and Methods" section.
There are also severe doubts that the surgical procedures were applied properly. Implants should be installed in such a way, that they are primarily located in the cortical bone of the tibia with its coronal part. Figure 1b clearly shows that no primary stability was achieved and the implant had slipped into the marrow space. Osseointegration is very different there from the regions directly adjacent to the cortical bone.
Figures 1a and 1b were obtained when studying the same limb of the same animal, as well as 2a and 2b.
Left unchanged.
The main result of the study as it is presented, appears to be the higher bone density at 3 days. At this time point usually no or very little bone formation has taken place. The dominating process at three days is resorption of fractured bone debris produced by the drilling process. The higher bone density in the treatment group is very likely a result of more bone debris present. That means less resorption? It hardly proves that better osseointegration has taken place as a result of treatment with extracellular vesicles. And if so why is there no higher bone density after 7 and 10 days?
We do not write that the increase in bone density 3 days after the operation was due to bone formation. Control and experimental animals were operated under the same conditions, by the same surgeon, therefore, the volume of bone detritus in all animals should be approximately the same. The increase in bone density, most likely, in our opinion, was due to the lesser severity of hyperemia and edema (relative area of ​​blood vessels in section, Table 1, 2), which, in turn, may be a consequence of a less significant inflammatory process, suppressed as a result of the immunomodulatory effect of MSC EVs. Bone debris is located more compactly, which gives a higher tissue density and creates conditions for the consolidation of bone fragments, but not their necrosis and lysis ("as a result of the immunomodulatory action of MSC EVs, the activity of inflammation decreases, and the bone fragments formed during implantation are mainly consolidated with each other and with the regenerating bone"). After 7-10 days, bone density did not decrease after the application of MSC EVs, but edema and hyperemia decreased in control animals, which statistically significantly increased bone density (Table 1).
Left unchanged.
The interpretation of the findings is by far too optimistic, appears to be one-sided, uncritical and biased in the direction of the hoped for results.
We found an increase in bone density after the application of MSC EVs and, of course, are trying to explain it as a result of the action of these MSC EVs. Of course, we are not completely sure of this, and everywhere we write "It is possible", "It is most likely", "probably", etc. And the manuscript is entitled "First Experimental Study ...."
Left unchanged.
I hope that we have answered all the questions and comments of the Dear Reviewer, and also clarified the controversial points of our manuscript.

Reviewer 2 Report
Authors tried to evaluate the effect of MSC EV on osteointegration with implants. Supplementation is needed to be paper.
Ct scan is needed for 3 D morphometric analysis.
The site of drilling are different. Far cortex is drilled in Fig.2 but not in Fig.1.
Distribution of inflammatory or osteogenic cells should be included for histopathological analysis.
For histomorphometric analysis on osteointegration quantification of bone, chondral and fibrous tissues around implants should be needed.
Author Response
I want to thank the Dear Reviewer for the short and clear review. On the substance of the remarks made:
Ct scan is needed for 3 D morphometric analysis.
Computer scans with 3-D modeling were made for each case in control and experimental animals (Figure 1b, 2b). But this was done not for morphometric analysis, but to confirm the correct placement of the screw implant and to measure the density of bone tissue next to the implanted device.
The site of drilling are different. Far cortex is drilled in Fig.2 but not in Fig.1.
The cortical layer of the opposite side of the tibia was not drilled in any case, but sometimes the screw implant rested against this bone plate. In this experiment, we did not measure the fixation strength of the implant, but only the bone density next to the foreign body was measured in various times after implantation.
Distribution of inflammatory or osteogenic cells should be included for histopathological analysis.
The results of the study of the numerical density and the percentage proportion of tissue leukocytes in soft tissues on proximal tibial condyles near the implants are added to the tables and the text of the manuscript.
For histomorphometric analysis on osteointegration quantification of bone, chondral and fibrous tissues around implants should be needed.
We did not observe the formation of cartilage; restoration of the damaged bone proceeded via the angiogenic pathway, not the enchondral one. To determine the osseointegration of the implants, the length of contact of the surface of the device directly with the bone or with the connective (fibrous) tissues was measured ("The length of the contact implant profile with the bone tissue was set as a percentage of the total length of the contact profile of the screw thread with the tissues; we the curvimeter used for that."). The results of measuring the length of contact of the surface of the implant with the bone tissue are shown in the tables and are expressed as a percentage of the entire length of the surface of the product, respectively, the rest of the surface of the implant is in contact with the connective or fibrous tissues.
Once again, I thank the Dear Reviewer for studying the manuscript, and I hope that we have answered all of his questions and comments.

Reviewer 3 Report
The aim of study is clear and of interest to readers, however several issues need to be addressed.
The introduction is short and lack important points. the authors should refer to previous use of EVs in bone healing generally and dental use specifically.
The methods section include enough details, but some methods are too long and can be divided into paragraghs.
The results are presented limitedly showing x-ray and histology images. The authors should expand this section with describing and showing figures for flow-cytometry images for MSCs and for EV characterisation. Importantly, there is no quantitative comparing between implant versus EVs. The authors are advised to use a quantified histological analysis or another tool to measure the difference as it is currently only descriptive.
The conclusions cannot be based on the current results.
Author Response
Many thanks to Dear Reviewer for the short and clear review. On the remarks, questions and comments made:
The introduction is short and lack important points. the authors should refer to previous use of EVs in bone healing generally and dental use specifically.
Fixed. The "Introduction" is expanded, in this section we have added the results of earlier studies of using MSC EVs for bone regeneration, for influencing inflammation and for healing bone defects made for dental implants.
The methods section include enough details, but some methods are too long and can be divided into paragraghs.
Fixed. Long paragraphs in "Materials and Methods" are separated.
The results are presented limitedly showing x-ray and histology images. The authors should expand this section with describing and showing figures for flow-cytometry images for MSCs and for EV characterisation. Importantly, there is no quantitative comparing between implant versus EVs. The authors are advised to use a quantified histological analysis or another tool to measure the difference as it is currently only descriptive.
This experiment compares the results of peri-implant bone regeneration without and after the use of MSC EVs. The results of studying the effect of MSC EVs on the regeneration of a bone defects without implants were carried out by us earlier (Maiborodin IV; Shevela AA; Marchukov SV; Morozov VV; Matveeva VA; Maiborodina VI; Novikov AM; Shevela AI Regeneration of the bone defect at experimental application of extracellular microvesicles from multipotent stromal cells. Novosti Khirurgii. 2020, 28, 359-369, doi: 10.18484/2305-0047.2020.4.359.), and these results are presented in the new edition of the "Introduction". A quantitative analysis of histological samples with statistical processing of the data obtained was made, the results are presented in tables 1 and 2 and in the "Results" section.
Images depicting the results of flow cytometry for characterizing MSC and EVs are generally known and do not contain any new information; the literature contents many such figures. But if the Dear Reviewer insists, we will include these images in the article, but not in the "Results" section, but in "Materials and Methods".
The conclusions cannot be based on the current results.
I will take the liberty of disagreeing with the Dear Reviewer. We found a fact - an increase in bone density next to a metal screw implant after using MSC EVs. This fact is reflected in the conclusions. We also tried to logically link this fact with the already known and described effects of MSC EVs. Of course, we are not completely sure about this, and everywhere we write "It is possible", "It is most likely", "probably", etc. And the manuscript is entitled "First Experimental Study ...."
Thanks again to the Dear Reviewer for the time spent reading our manuscript and writing a review.

Round 2
Reviewer 1 Report
The authors have substantially improved and altered the manuscript according to the reviews content. The introduction, methodology and discussion has been amended with additional information and reasoning.
Although not completely convincing, they were able to explain why their chosen methodology is adequate (paraffin histology). They have also rebutted my suspicion that the implants were not placed correctly in the cortical bone by showing images from a different viewing angle that prove me wrong. They also bring some justified arguments to defend their somewhat optimistic and biased interpretation of the results.
The weakest point of the manuscript still remains the use of the English language. Some post acceptance editing would be advisable.
Author Response
I thank Dear Reviewer. Regarding the comments on the quality of the translation, I can only promise to make an attempt to change the translation agency.
Reviewer 2 Report
As the reason that bone density of EV group was significantly higher than without EV authors suggested and concluded as follows; It is possible that as a result of the immunomodulatory action of MSC EVs, the activity of inflammation decreases, and the bone fragments formed during implantation are mainly consolidated with each other and with the regenerating bone. --> not scientific. Show the data and Discuss about inflammatory cells, osteoclast and osteoblast or their markers in contact area near the implant, not in soft tissue covered implant and bones near it
Author Response
Thanks again for the job.
Leukocyte infiltration in bone tissue and in tissues immediately adjacent to the implant has not been investigated, since long-term decalcification strongly changes the morphology of cells, and the work was based on morphological data only.
This note has been added to the "Materials and Methods" section.
The Discussion says: "As a result of the MSC EV introduction, the activity of inflammation decreases [16,17] and, respectively, the degree of vasodilation is lower. ... can be noted that the use of MSC EVs led to a smaller number of leukocytes in the soft tissues from the surface of the proximal tibial condyle already on the 3rd day after implantation and to a less pronounced decrease in their number further, in comparison with the surgery performed in the control group. These results are best explained from the standpoint of the well-known immunomodulatory effect of MSC EVs [15,16,17], which results in suppression of lymphocyte [15] and neutrophil [16,17,34] activity, and the severity of inflammation is also suppressed [16,17]". We believe that this just justifies the relationship between changes in vascularization (less severity of hyperemia) and inflammatory infiltration (fewer lymphocytes and neutrophils) with the immunomodulatory effect of MSC EVs.
Based on the comments of the Dear Reviewer, in the "Conclusion" added rationale for the relationship between changes in vascularization and leukocyte infiltration and suppression of inflammation.
Reviewer 3 Report
The authors have edited the manuscript according to the comments. However, some results in tables should be presented in figures (e.g. bars) and the numbers of replicates (animal samples) should be clarified. Also, how many studies have extracted MSC-EVs from rabbit dental tissues? I assume not many, therefore, I still find it important to show flow-cytometry plots for characterisation of these EVs.
Author Response
Thanks again.
The number of animals for each period in each experimental group was added to the "Materials and Methods" section.
Based on the remarks of the Dear Reviewer, and, despite the basic requirements on the inadmissibility of duplicating the materials of tables and graphs, graphs (bars) based on data from tables have been added to the manuscript.
I am somewhat surprised by the Reviewer's remark that "... extracted MSC-EVs from rabbit dental tissues". Nowhere in the manuscript does it say that we used MSC from rabbit teeth to obtain EVs. Flow cytometry plots will be attached to the manuscript as Supplements (Figure S1).
|
ISO |
|
CD90 |
|
CD106 |
|
CD45 |
|
ISO |
Figure S1. The phenotype of rat mesenchymal stem cells (MSCs) derived from the bone-marrow. Data indicate flow cytometry analysis. The MSC of rat were stained with specific antibodies or isotype controls. The samples were analyzed on a flow cytometer "FACSAria" III using FACSDiVa Version 6.1.3. software (Becton Dickinson).
